# Symptomatic Infection with *Vairimorpha* spp. Decreases Diapause Survival in a Wild Bumble Bee Species (*Bombus griseocollis*)

**DOI:** 10.3390/ani13101656

**Published:** 2023-05-17

**Authors:** Margarita Orlova, Monique Porter, Heather M. Hines, Etya Amsalem

**Affiliations:** 1Department of Entomology, The Pennsylvania State University, University Park, PA 16802, USA; 2Department of Biology and Chemistry, College of Arts and Sciences, State University of New York Polytechnic Institute, Utica, NY 13502, USA; 3Department of Biology, The Pennsylvania State University, University Park, PA 16802, USA

**Keywords:** diapause, *Vairimorpha*, bumblebee, nutrition, parasitism

## Abstract

**Simple Summary:**

Our work shows that bumble bee queens that grew up in a colony infested by parasites are less likely to survive the winter diapause. However, infection levels in overwintering queens have little impact on survival length. The factor that impacts the survival length of infected queens the most is their body size, reflected in both body mass and head capsule size. Larger queens from infected colonies survived for longer periods than smaller queens, whereas no such effect was observed in queens from healthy colonies. The variation in body size among queens who grew up in infected colonies was much larger than among those from healthy colonies, suggesting inconsistency in nutrition availability in the maternal colony. Overall, our findings suggest that parasite infection can have indirect detrimental effects on diapause survival that can be mediated via nutrition.

**Abstract:**

*Vairimorpha*, a microsporidian parasite (previously classified as *Nosema*), has been implicated in the decline of wild bumble bee species in North America. Previous studies examining its influence on colony performance have displayed variable results, from extremely detrimental effects to no observable influence, and little is known about the effects it has on individuals during the winter diapause, a bottleneck for survival in many annual pollinators. Here, we examined the effect of *Vairimorpha* infection, body size, and mass on diapause survival in *Bombus griseocollis* gynes. We demonstrate that gyne survival length in diapause is negatively affected by symptomatic *Vairimorpha* infection of the maternal colony but does not correlate with individual pathogen load. Our findings further indicate that increased body mass offers a protective effect against mortality during diapause in infected, but not in healthy, gynes. This suggests that access to adequate nutritional resources prior to diapause might offset the harmful effect of *Vairimorpha* infection.

## 1. Introduction

Decline in the numbers of wild bees is an alarming ecological trend that has been well documented in the last few decades [1,2] and has been attributed to many factors. Among these factors are habitat loss, climate change, use of pesticides, decrease in resource availability, pathogens, and parasites, particularly viruses, ectoparasitic mites, and unicellular group-specific parasites such as *Crithidia bombi* and *Vairimorpha bombi* [3,4,5]. Large-scale surveys of pathogen levels in bumble bee species have implicated *V. bombi* as a leading factor in the decline of the most threatened species in North America [6,7,8]. *V. bombi* is one of the several species of the microsporidian genus *Vairimorpha* that were formerly assigned to the genus *Nosema*. Recent molecular data, however, firmly separated the two genera and reassigned several species parasitizing insects to *Vairimorpha* [9]. The microsporidian genus *Vairimorpha* includes multiple species that can infect diverse insect and non-insect hosts [10]. In bees, these are often host-specific (e.g., *V. bombi* in bumble bees), but some, like *V. ceranae*, can move between host species, making them more broadly threatening for wild bees [11]. Recent studies also suggest that certain species of *Vairimorpha* can infect hosts belonging not only to different species but to different genera (Jones et al., 2022). 

The genus *Vairimorpha* has a common mode of function across insects, whereby it proliferates in the gut epithelium and is released into the gut lumen, from which it is excreted with feces and subsequently infects other individuals. Effects of *Vairimorpha* infection on host colony performance have been documented in numerous bumble bee species and were shown to vary across studies. In both *B. terrestris* and *B. occidentalis,* studies showed no negative effects of *V. bombi* infection on colony productivity parameters such as population growth and colony weight [12,13]. Other studies in *B. terrestris* demonstrate that, in combination with other stressors, *V. bombi* infection can reduce brood survival [14] and can severely reduce colony productivity [15] and the individual fitness of queens and males [16]. Bumble bees can transmit *V. bombi* among communities via adults [17]; however, infection within the colony occurs mostly at the larval stage. Larvae are apparently more easily infected than adults, and adult bees developing from infected larvae can transmit the disease to other larvae, perpetuating the infection at the colony level [18].

Bumble bees and most wild solitary bees that provide pollination services are annual and undergo winter diapause before they initiate a colony or a nest. Bumble bee gynes (females that mate and become queens) emerge in late summer, forage to acquire nutritional resources, mate, and spend the winter in diapause until the following spring, when they initiate a colony that becomes social once the first worker emerges [19,20,21]. The winter diapause is thought to be a period of high vulnerability, and changes in individual fitness during this period may have disproportional impacts on colony initiation and post-diapause performance [22,23,24,25]. 

Previous studies in bees have demonstrated the influence of factors such as body mass and nutrient sequestration prior to diapause on the outcome of diapause. They show that the buildup of lipids, proteins, and glycogen reserves in early adulthood is crucial for fueling metabolism during diapause, surviving diapause, and also post-diapause reproductive performance [25,26,27,28,29,30]. Additionally, high concentrations of cryoprotectants such as glycerol in the hemolymph can be protective against low temperatures and are also acquired prior to diapause onset [25,31]. The acquisition and storage of nutrients depend not only on floral availability and feeding but also on their interaction with other factors, such as exposure to pathogens. Studies performed mainly in the honey bee demonstrate that a high-quality diet, rich in macronutrients such as proteins and amino acids and micronutrients such as iron and calcium, can have a protective effect against the harmful effects of pathogen infection and increase the insects’ tolerance against infectious agents [32,33,34]. On the other hand, infection, especially with gut parasites, can impede the digestion of food and sequestration of nutrients and alter energy metabolism [35,36,37], which may also affect diapause success. 

While all pathogen infections can cause stress and negatively affect nutrition at the colony level, the impact of *Vairimorpha* infection on nutritional state can be especially significant given that the parasite attacks the gut tissue of the host, causing a disruption in digestion and nutrient absorption. The parasites penetrate the intestinal epithelium, and their life cycle proceeds within the host epithelial cells. The sporoplasm injected into the cell produces meronts, small diplokaryotic cells that undergo multiple rapid divisions. The cells resulting from these divisions morph into larger diplokaryotic cells with a thick cell wall, termed sporonts. These, in turn, undergo further divisions to produce the infective spores. It is at this stage that the host cell is disrupted and is releasing spores into the gut lumen, from which they are excreted with feces and proceed to infect other individuals. 

Our study examines the influence of *Vairimorpha* spp. infection on diapause survival in gynes of the North American bumble bee species *Bombus griseocollis*. Gynes collected from colonies with and without *Vairimorpha* spp. symptoms were placed in cold storage conditions simulating diapause. Survival duration, ovary activation, and metrics of body mass and size were determined. These metrics were further correlated with individual *Vairimorpha* spp. load in the gynes’ gut upon their death. We hypothesize that both colony and individual levels of *Vairimorpha* spp. will impair diapause success and that heavier and larger individuals will show higher resilience to the presence of the pathogen.

## 2. Methods

### 2.1. Bees’ Maintenance

Founder queens of *Bombus griseocollis* were collected in the wild from several sites in the region around State College, Pennsylvania, USA (Villalona et al., 2020). Colonies from these queens were reared in the lab in an incubator (60–80% humidity, 26 °C) during the summer of 2019 and were individually fed from the same original sources of *ad libitum* 50–60% sugar solution or Biobest Biogluc and fresh frozen pollen collected by honey bees (Swarmbustin’ Honey, West Grove, PA, USA). By the end of the summer, 36 gynes were collected from four colonies (7–11 gynes per colony). Two of the colonies (*n* = 16 gynes) were suspected to be infected with *Vairimorpha* spp., evidenced by liquid bee feces covering the colony floor, while the other two colonies did not display these signs (*n* = 20 gynes). 

### 2.2. Diapause Simulation

Gynes obtained from their natal colonies were placed in cold storage conditions (4 °C and >90% humidity), simulating winter diapause as described in [25]. Bumble bee diapause length varies across species, ranging between 6 and 9 months in the wild [19], and is comparable to the length of survival in cold storage in the lab [25]. Humidity levels were achieved by placing bowls of water in the fridge. All gynes from one colony were placed in a 20 cm× 30 cm× 3 cm cardboard box with openings for ventilation. Examination of diapause survival started one week after placement in cold storage and continued until the last gyne had died, for a total of 23 weeks. Survival was assessed once a week by placing the cardboard boxes at room temperature (22 °C) and under ambient light for 10 min. During this period, the gynes were observed for movements. Gynes that displayed movement were recorded as alive and placed back in cold storage. Gynes that failed to move within 10 min were recorded as dead. Dead gynes were placed at −80 °C until further analyses, and their date of demise was recorded. 

### 2.3. Weight and Body Size Measurement

Dead gynes were weighed using an analytical scale, and the width of the head capsule was measured using an eyepiece micrometer under a stereomicroscope [38,39]. Mass was measured in all 36 gynes, whereas head capsule was measured in 24 gynes (11 in colonies displaying *Vairimorpha* symptoms and 13 in control colonies). In the additional 12 gynes, the head was destroyed during dissection, and its size could not be measured.

### 2.4. Dissection of Gut and Ovaries 

All gynes were dissected under a stereomicroscope. Ovaries were excised and placed in a separate drop of water for measurement, while the gut was placed on a sterile plate where it was cut lengthwise in two with a sterile blade. One half of the gut was placed in a sterile 1.7 mL tube with 100 µL molecular grade water and zirconia beads for subsequent homogenization and staining, and the rest was used for other applications. Dissections of gynes from symptomatic colonies revealed parasite-induced damage to internal organs and especially the ovaries, similar to what is described in (Otti and Schmid-Hempel, 2007) (12 out of 16 gynes), and therefore the measurements of ovary size are not reported. 

### 2.5. Quantification of Vairimorpha

Gut samples were homogenized in a fast-prep machine at maximum speed for 45 s. The procedure was repeated twice to disrupt tissue and obtain a homogeneous suspension. Five µL of the suspension was smeared on a glass slide over an area of ca. 2 cm × 2 cm. The wet smears were briefly inspected under the microscope before fixation. Wet smears were fixed for 3 min in methanol (Sigma-Aldrich, St. Louis, MO, USA), washed for 1 min in staining buffer solution (Sigma), and placed into a staining jar with Giemsa stain solution (Sigma) for 20 min. Staining buffer and Giemsa stain solution were prepared according to manufacturer instructions. After staining, the slides were washed twice in staining buffer solution for 1 min each time and air dried. Stained slides were inspected under the microscope and photographed using a Nikon DS-Fi3 camera. Four different random areas on each slide were photographed. 

The life cycle of *Vairimorpha* within the host progresses through several stages involving three different cell types. The sporoplasm injected into the cell produces meronts—small diplokaryotic cells that undergo multiple rapid divisions. The cells resulting from these divisions morph into larger diplokaryotic cells with double nuclei and a thick wall, termed sporonts. These, in turn, undergo further divisions to produce the infective spores. In the current study, the cells observed in gut preparations were sporonts. *Vairimorpha* sporonts appearing in each photograph were counted and averaged. The average number of sporonts per field of view was used in further analysis. Identification of *Vairimorpha* sporonts was performed by comparison with literature (e.g., [40] and with smears made from highly concentrated samples of *Vairimorpha* suspension from a previous experiment. 

### 2.6. Statistical Analysis

All statistical analyses were performed using IBM SPSS v.21. Cox regression survival analysis was used to assess the effect of categorical variables, including symptomatic infection and colony identity (i.e., which colony the gyne originated from), on the length of diapause survival. Generalized linear mixed model analysis (GLMM) was performed to assess the covariance between continuous variables, such as number of sporonts in smears and body mass. Robust estimation was used to handle violations of model assumptions, and Satterthwaite correction was employed to account for small and unequal sample sizes. Generalized linear mixed model analysis was performed on standardized values (Z-scores) to obtain standardized beta coefficients. Non-parametric correlation (Spearman’s ρ) was used as a conservative estimate to assess the relationships between continuous variables. Statistical significance was accepted at α = 0.05. Data are presented as means ± SE.

## 3. Results

Gynes from control colonies without symptoms exhibited, on average, nearly zero count of *Vairimorpha* sporonts (0.92 ± 0.3, mean ± SE) compared to gynes from colonies that exhibited symptoms (29.2 ± 6.8) (GLMM, F_1,32_ = 18.31, *p* < 0.001) (Figure 1A). However, the number of sporonts in gynes from symptomatic colonies varied substantially (3–80 sporonts per field of view). The identity of the natal colony did not impact the number of sporonts (F_2,30_ = 0.886, *p* = 0.42). 

The presence of symptoms in the maternal colony significantly affected the length of diapause survival and was lower in gynes from symptomatic compared to control colonies (Cox regression, overall χ^2^_2_= 8.55, *p* = 0.014, Wald χ^2^_1_ = 6.91, *p* = 0.008 for symptoms, Wald χ^2^_1_ = 3.71, *p* = 0.054 for colony identity). No correlation was found between the length of survival in diapause and the individual sporont count in all gynes (Spearman’s correlation: ρ = −0.16, *p* = 0.35, *n* = 36), and the same was found when the gynes from symptomatic colonies were analyzed separately (Spearman’s ρ = 0.25, *p* = 0.35, *n* = 16). The data for the length of gyne survival and its correlation with sporont counts is presented in Figure 2A,B, respectively. As mentioned in the Methods section, 12 out of 16 gynes from symptomatic colonies (75%) had significant damage to their ovaries and other internal organs. However, the presence of damage to internal organs was not a predictor of survival length in these gynes (Cox regression, overall χ^2^_2_= 9.67, *p* = 0.008, Wald χ^2^_1_ = 3.34, *p* = 0.07 for damage, Wald χ^2^_1_ = 6.94, *p* = 0.011 for colony identity). 

The average body mass and head capsule width correlated significantly (Spearman’s ρ = 0.74, *p* < 0.0001) and did not differ significantly between the gynes from symptomatic (0.76 ± 0.02 g and 5.3 ± 0.05 mm, respectively) and non-symptomatic colonies (0.79 ± 0.04 g and 5.3 ± 0.07 mm, respectively); however, in the gynes from diseased colonies, gyne body mass was more variable than in healthy ones (Levene’s W = 5.03, *p* = 0.03), and the average body mass significantly differed between the two symptomatic colonies but not between the two non-symptomatic ones (GLMM, F_1,30_ = 17.1, *p* < 0.0001 for symptomatic colonies; F_1,30_ = 0.27, *p* = 0.6 for non-symptomatic colonies). Neither body mass nor head capsule width covaried with the average number of sporonts per field of view (GLMM, F_1,28_ < 1, *p* > 0.3 for all analyses). Both mass and size correlated with the length of diapause survival in gynes from symptomatic colonies (body mass: Spearman’s ρ_16_ = 0.57, *p* = 0.02; head capsule width: Spearman’s ρ_11_ = 0.61, *p* = 0.02) but not in gynes from control colonies (body mass: Spearman’s ρ_18_ = −0.33, *p* = 0.17; head capsule width: Spearman’s ρ_13_ = −0.36, *p* = 0.21) (Figure 3). A GLMM analysis found that the body mass of gynes from infected colonies significantly predicted survival length but sporont count did not (GLMM, F_1,13_ = 12.16, *p* = 0.002 for body mass; F_1,12_ = 2.07, *p* = 0.12 for the average number of sporonts per field of view), while in gynes from healthy colonies neither was a significant predictor for diapause survival (GLMM, F_1,15_ < 2, *p* > 0.1 for both parameters). 

## 4. Discussion

Our results indicate that symptomatic infection with *Vairimorpha* in the maternal colony has a significant negative effect on the length of diapause survival of *Bombus griseocollis* gynes. On average, gynes from symptomatic colonies had higher sporont counts in their gut tissue than gynes from colonies without symptoms, which had no to low levels of sporonts. However, sporont counts per gyne by themselves did not predict the length of diapause survival. Below, we discuss two potential explanations for this phenomenon. We further found that higher body mass at the onset of diapause protects gynes infected with *Vairimorpha* from mortality but does not correlate with survival length in healthy gynes. Below, we also discuss potential explanations and the role of body mass during diapause. 

As expected, the symptoms of *Vairimorpha* matched with the average load of *Vairimorpha* in gynes from symptomatic and non-symptomatic colonies. Post-mortem dissection indicated a damage to internal organs in 75% of gynes from symptomatic colonies, but no gynes from non-symptomatic colonies showed such damage. Similar effects of Vairimorpha infection are described in another bumble bee species and in the honey bee (Otti and Schmid-Hempel, 2007, Hassanein, 1951). Gynes from symptomatic colonies also survived for a shorter length of time in diapause. This may indicate that infection in itself impairs diapause survival and therefore the individual load should also negatively correlate with the length of survival, but this was not the case. Parameters related to infection such as individual sporont load or damage inflicted by the parasite on internal organs were not predictors of diapause survival length. However, survival length was significantly associated with metrics of body size—body mass and head capsule width. These findings suggest that the effect of maternal colony infection on gyne survival in diapause is explained by the indirect effects of colony-level infection on brood development. In this case, the presence of *Vairimorpha* by itself may not be detrimental during diapause. Instead, developing in a colony which was heavily infected with *Vairimorpha* may affect the nutrition available for gynes during development or early adulthood, when they acquire the macronutrients that are crucial for their diapause survival [25]. While we cannot attest to the timepoint at which the colony acquired the parasite, we can hypothesize that an early exposure could affect the available nutrition in the colony during gyne development and eclosion, the overall foraging efforts of workers, the colony mass, the quality of care the brood received, or the ability to digest or absorb important macronutrients during a critical timepoint in the life cycle. It is important to note though that while there were no differences in the average mass or head size of gynes from symptomatic and healthy colonies, there was significant variability within and between the two symptomatic colonies, and the colony with larger and heavier gynes outperformed the colony with smaller and lighter gynes in terms of diapause survival. Increased variability in size and mass among gynes reared in symptomatic colonies suggests inconsistency in nutrition availability in such colonies, probably occasioned by the infection. 

The average metrics of body mass and size were comparable between gynes from diseased and healthy colonies; however, body mass was more variable in gynes from diseased colonies than in healthy ones, and this variation was manifest both between and within diseased colonies. Notably, sporont counts did not correlate with body mass. This finding suggests either that effects of *Vairimorpha* infection on queen physiology are inconsistent and multiple factors are at play determining the relationship between body mass and infection, or that infection plays no role in the determination of body mass, but body mass can affect how the organism interacts with the pathogen. Interestingly, the literature suggests that in *Bombus* and *Osmia*, the infection with *Vairimorpha* has no effect on body mass [13,41,42], while data from the honey bee suggests that infection with *N. ceranae* induces lipid loss [43]. It is important to mention that since in this study mass was measured after diapause, it is difficult to rule out the loss of fat during this period. However, we observed an almost perfect correlation between body mass (which is determined in early adulthood) and head capsule width (which is determined during development), and it would be safe to say that larger gynes were also generally heavier and their mass is probably determined by the same developmental factors as their size. 

While body size and mass did not differ between the groups, body size and mass did have a protective effect on survival in diseased gynes. Body size and mass can be a reliable proxy for energetic reserves [44,45,46]. A protective effect of body size against adverse temperature conditions during diapause has been previously documented in various *Bombus* species [22,47,48]. Reaching a critical body mass prior to diapause is also detrimental for surviving diapause [22,24,25,49]. Our finding that the protective effect of large body size on diapause survival exists only in gynes from diseased colonies suggests that the harmful effect of infection on survival is probably mediated via increased energy expenditure, and large and heavy gynes possess an initial energetic advantage over small and light ones. *B. griseocollis* produces especially large queens, and it has been observed previously that diapause is compromised most in queens falling below a certain smaller size threshold [22,50], which could explain the lack of effect of body size in healthy colonies. Sick colonies produced a few queens of smaller size that also had low duration of diapause survival; however, these alone do not explain the correlation in this group.

One obvious way of *Vairimorpha* affecting the energy balance is via damage to the gut and disruption of nutrition, as well as direct energy consumption by the parasite cells. This effect, while very plausible, is difficult to quantify, and further study is needed to assess how much energy is lost through gut disruption and how much energy parasite cells consume directly. Another possible avenue through which *Vairimorpha* infection might affect diapause survival is via the energetic effects of immune challenge. Previous studies in bees have shown that immune response is energetically costly and triggers increased consumption of food. On the other hand, it has been shown that limited nutrition can negatively affect immune function [51,52,53]. This suggests that heavier queens are better able to sustain the energetic cost of immune response. 

Certainly, *Vairimorpha* infection can affect the use of energy and the metabolism of its host both directly and indirectly through damaging the gut tissue and causing diarrhea [54], but other, subtler effects can also play a role: *Vairimorpha* parasites can increase juvenile hormone levels in the host, which, in turn, can alter the host’s metabolic processes [55,56]. However, the finding that body size is a better predictor of diapause survival than infection intensity per se raises the question as to how important these direct effects are compared to the indirect colony-level effects mediated by conditions in the maternal colony. Our study emphasizes the necessity of further research to disentangle the individual and colony-level effects of pathogens and parasites and to understand possible mechanisms through which these effects are mediated. 

It is, however, important to mention several caveats that might affect the interpretation of our results. The first is the low number of colonies reared from wild queens. While the four colonies in our study yielded a sufficient number of queens to observe trends in diapause survival and achieve meaningful conclusions, a larger number of colonies would allow better assessment of the impact of parasite infestation on queen rearing dynamics. Secondly, while we operated on the assumption that wild bumble bees were carrying *V. bombi*, this has not been confirmed using molecular data. Indeed, a recent study suggests that wild bees (including bumble bees) in the northeastern US are commonly affected by *V. apis,* which apparently occurs in a wider range of hosts than previously expected [57]. In addition, while we observed symptoms consistent with intestinal infection and found no signs of infection by other intestinal parasites beside Vairimorpha, it is important to mention that hosts of Vairimorpha might be co-infected with bacterial or viral pathogens that are not readily observed by microscopy. Additional work on the probability of such coinfection and its effects on gyne physiology might be needed. Finally, it is important to note that in our study the longest survival of queens in conditions simulating diapause (23 weeks) was still much shorter than that reported for other species [22,25]. For example, under the same conditions, *B. impatiens* gynes survived up to 40 weeks, which is equivalent to their length of diapause in the wild [19]. This difference might be attributed to differences in species biology. Little is known about the natural history of bumble bee species that are not commercially available, and this gap is critical to bridge in order to predict how pathogens alone and in concert with other stressors (e.g., pesticides, global warming, and reduced floral availability) might affect diapause survival and pollination services. 

Overall, our study demonstrates the impact of a common pathogen on diapause survival in an important pollinator. We show that colony-level symptoms are a stronger predictor of diapause survival in gynes than the individual load of the *Vairimorpha* and that body mass may protect gynes from the detrimental impact of the pathogens. 

## 5. Conclusions

Our results suggest that growing up in a colony with symptomatic Vairimorpha infection has detrimental effects on diapause survival in bumble bee queens. However, these effects are not directly related to individual parasite load but are dependent on the queens’ body size. The detrimental influence of the parasite is likely mediated by rearing conditions in the infected colony. 

## Figures and Tables

**Figure 1 animals-13-01656-f001:**
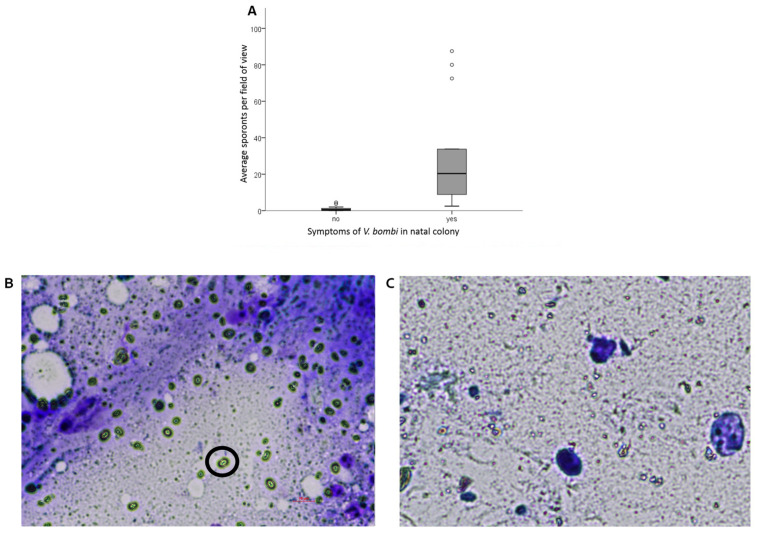
The prevalence of *Vairimorpha* sporonts in the gut smears of gynes from symptomatic and non-symptomatic colonies. (**A**) The average number of sporonts per field of view (average of 4 fields of view). Data are presented as boxplots displaying the median value as a line and 25–75 quartiles with dots above each box indicating outliers. (**B**) An image of a gut smear of a heavily infected gyne from a symptomatic colony. A representative sporont is indicated by a black circle. Scale bar = 10 µm. (**C**) An image of a gut smear of a gyne with no infection of *Vairimorpha* from a healthy colony.

**Figure 2 animals-13-01656-f002:**
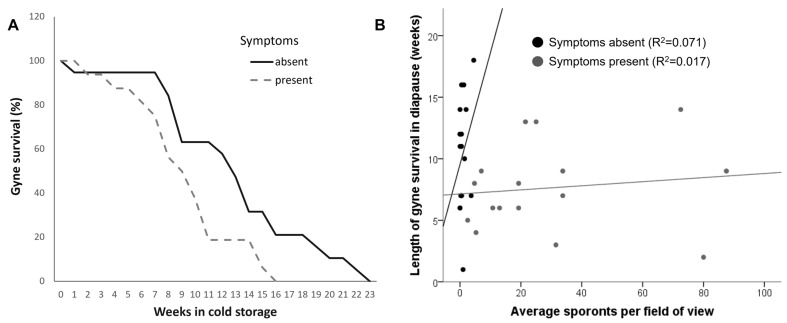
Survival of gynes from colonies with and without symptoms of *Vairimorpha bombi* infection in cold storage. (**A**) Percentage of surviving gynes from symptomatic colonies (*n* = 16 gynes) and healthy colonies (*n* = 20 gynes). Gynes were monitored for survival weekly until the last gyne had died. (**B**) Length of survival in cold storage of gynes from symptomatic (*n* = 16 gynes) and healthy (*n* = 20 gynes) colonies as a function of the average sporont counts in gut smears. R^2^ values are based on a linear fit.

**Figure 3 animals-13-01656-f003:**
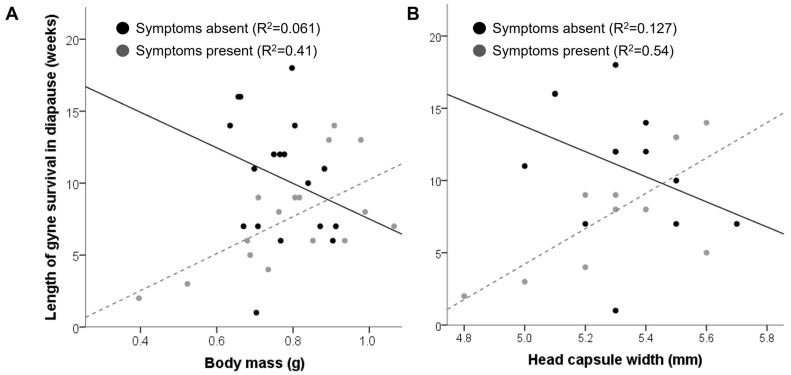
The relationship between body size and mass and cold storage survival in gynes from symptomatic and healthy colonies. (**A**) Cold storage survival as a function of body mass (*n* = 36). (**B**) Cold storage survival as a function of head capsule width (*n* = 24). R^2^ values are based on a linear fit.

## Data Availability

The data presented in this study are available on request from the corresponding author.

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
