# Peer review of "Symptomatic Infection with Vairimorpha spp. Decreases Diapause Survival in a Wild Bumble Bee Species (Bombus griseocollis)"

_animals, 2023, doi:10.3390/ani13101656_

Round 1

Reviewer 1 Report

This manuscript presents an interesting topic on bumble bee diapause, It was written smoothly, and statistic assay are well presented.  however, there are some questions need to be clarified:

L175-176:  symptomatic in fection symptomatic infection redundant words?

L176: Could you please clarify colony identity for readers convenience.

L151-160: In 2.5 Quantification of V. bombi, Spore counting on hemocytometer is more widely used, easier than the method described in the manuscript, which may cause the data more variable (L189). Why did the authors choose the complicated one?

L333-340: As the authors mentioned that the pathogen species has not been confirmed by molecular data, however this could not be skipped if the author want to name this microsporidia as V. bombi, rather than V. apis.  

Futhermore, as these  gynes were collected from wild, they may be infected with other pathogens as well, how to rule out their influences on the survival?

Author Response

Reviewer #1

This manuscript presents an interesting topic on bumble bee diapause, It was written smoothly, and statistic assay are well presented.  however, there are some questions need to be clarified:

L175-176: “ symptomatic in fection symptomatic infection” redundant words?

Thanks for pointing this out. The redundancy has been corrected

L176: Could you please clarify “colony identity” for readers convenience.

Thanks for this suggestion, the clarification of the term “colony identity” has been added to line 176.

L151-160: In 2.5 ”Quantification of V. bombi”, Spore counting on hemocytometer is more widely used, easier than the method described in the manuscript, which may cause the data more variable (L189). Why did the authors choose the complicated one?

Thanks for this suggestion. While we are aware that counting on a hemocytometer is an easier method of counting spores, unfortunately, we do not have access to such equipment and alspo wish to preserve our samples on  microscope slides for future uses, e.g., as educational material

L333-340: As the authors mentioned that the pathogen species has not been confirmed by molecular data, however this could not be skipped if the author want to name this microsporidia as V. bombi, rather than V. apis.  

Thanks for this suggestion, the naming has been corrected to Vairimorpha spp., throughout the manuscript. We also added information in the introduction concerning the expanding host range of Vairimorpha parasites.

Futhermore, as these gynes were collected from wild, they may be infected with other pathogens as well, how to rule out their influences on the survival?

Thanks for raising this point. While we cannot rule out the possibility of infection with bacterial and viral pathogens, our dissections and analysis of gut samples showed no presence of Sphaerularia bombi or Crithidia bombi. We also added a brief discussion of this issue to the discussion section.

Reviewer 2 Report

The manuscript is interesting and easy to read it. The results were interesting. The must ve care with the use of Nosema (N.) and Vairimorpha (V.). Please check and unify nomenclature in Vairimorpha.

Author Response

Reviewer #2

The manuscript is interesting and easy to read it. The results were interesting. The must ve care with the use of Nosema (N.) and Vairimorpha (V.). Please check and unify nomenclature in Vairimorpha.

Thanks for this suggestion. Necessary corrections were made and we added a reference in the introduction on recent changes in taxonomic status of Nosema and Vairimorpha genera

Reviewer 3 Report

The manuscript entitled “Symptomatic infection with Vairimorpha bombi decreases diapause survival in a wild bumble bee species (Bombus griseocollis)” is very interesting and gives some new information about the infection Nosema bombi (Vairimorpha)and bumblebees. The experimental design is well presented. The work is generally correctly constructed. The figures are relevant to the content of the article, and they are appropriately described. The discussion was well written. The literature was selected accordingly. The paper is written in a very communicative way and, in my opinion, is acceptable after minor revision in the Animals.

Minor comment

If you want to use the term Vairimorpha instead of Nosema in the manuscript, give the exact source and briefly explain the reasons for changing the name of Nosema bombi to Vairiomorpha bombi.

Author Response

The manuscript entitled “Symptomatic infection with Vairimorpha bombi decreases diapause survival in a wild bumble bee species (Bombus griseocollis)” is very interesting and gives some new information about the infection Nosema bombi (Vairimorpha)and bumblebees. The experimental design is well presented. The work is generally correctly constructed. The figures are relevant to the content of the article, and they are appropriately described. The discussion was well written. The literature was selected accordingly. The paper is written in a very communicative way and, in my opinion, is acceptable after minor revision in the Animals.

Minor comment

If you want to use the term Vairimorpha instead of Nosema in the manuscript, give the exact source and briefly explain the reasons for changing the name of Nosema bombi to Vairiomorpha bombi.

Thanks for this suggestion. Necessary corrections were made and added a reference in the introduction on recent changes in taxonomic status of Nosema and Vairimorpha genera

Round 2

Reviewer 1 Report

1. The name of this microsporidia hasn't been corrected as Vairimorpha spp in the title and abstract.

2. As the manuscript concluded that Vairimorpha infection in gynes didnt significantly decreased the diapause survival rather then body size. The original title does not pinpoint  the target. is it possible revise it ? Here is my two cents: The maternal body size matters: Diapause survival depends on the gynes body sizes under Vairimorpha spp. infection in a wild bumble bee species (Bombus griseocollis)”.

Author Response

  1. Thanks for pointing this out, the nomenclature has now been checked and corrected throughout the manuscript
  2. We disagree with this statement, as we did, in fact, observe a decrease in survival in gynes from infected colonies, and body size in these gyne offered a protective effect that was absent in gynes from healthy colonies.